# AbFlex: Predicting the conformational flexibility of antibody CDRs

**Fabian C. Spoendlin** [1]   **Wing K. Wong** [2]   **Guy Georges** [2]   **Alexander Bujotzek** [2]   **Charlotte M. Deane** [1]

## Abstract

Proteins are highly flexible macromolecules and the ability to adapt their shape is fundamental to many functional properties. While a single, 'static' protein structure can be predicted at high accuracy, current methods are severely limited at predicting structural flexibility. A major factor limiting such predictions is the scarcity of suitable training data. Here, we focus on the functionally important antibody CDRs and related loop motifs. We implement a strategy to create a large dataset of evidence for conformational flexibility and develop AbFlex, a method able to predict CDR flexibility with high accuracy.

## 1. Introduction

Proteins are built of a sequence of basic building blocks (amino acids), which fold into a 3-dimensional shape. Many proteins adopt more than one stable structure, termed conformations, and transitions between them are fundamental for functional properties (Teilum et al., 2009). For example, flexibility plays an important role in antibodies, a class of proteins central to the immune system and frequently used for therapeutic drugs. Antibodies interact with their target through loops called CDRs. Conformational flexibility of CDRs has been linked to properties such as affinity (Mikolajek et al., 2022) and polyspecificity (Guthmiller et al., 2020; James et al., 2003). As both of these functional properties need to be optimised in therapeutic antibodies, data on CDR flexibility improve understanding of antibody function and enhance the drug development process.

Prediction of a single, 'static' structure of a protein from its sequence is now possible at high accuracy with a range of recently developed methods (Jumper et al., 2021; Baek et al., 2021; Lin et al., 2023). In contrast, machine learning tools to predict structures of conformational ensembles or indicate the ability of a protein to undergo conformational rearrangements are currently limited. A major challenge which has restricted progress in this field of research is the scarcity of suitable data for training and evaluation.

A number of methods have been developed in an attempt to predict structures of protein conformational ensembles. Most of these build on modifications of AlphaFold2 (AF2) to reduce the depth of the input multiple sequence alignment (MSA) (del Alamo et al., 2022; Stein & Mchaourab, 2022; Wayment-Steele et al., 2024; Sala et al., 2023b). These workflows increase the diversity of outputs during inference. A smaller number of generative models were developed specifically for the task of conformation prediction (Jing et al., 2023; 2024; Zheng et al., 2023; Lu et al., 2023; Mansoor et al., 2024). Methods have mostly been evaluated on one or a few case studies, which may not accurately reflect their performance across diverse sets of proteins (Riccabona et al., 2024; Faezov & Dunbrack, 2023; Sala et al., 2023a; Saldaño et al., 2022). A study evaluating performance on a slightly larger dataset of 100 proteins showed that methods generally increase diversity in predicted structures but do not capture the conformational landscape well (Jing et al., 2024). Correlations between predicted flexibility and observed flexibility in crystal structures appear to be weak (Jing et al., 2023). Stronger correlations were identified between predicted flexibility and flexibility in molecular dynamics (MD) simulations for models fine-tuned on MD data (Jing et al., 2024).

In this work, we develop a strategy to overcome issues of data scarcity. On the one hand, we focus on functionally important secondary structure motifs, antibody CDRs and related loops, rather than full length proteins. On the other hand, we implement a systematic approach of mining the protein data bank (PDB) (Berman et al., 2002). In this way, we create a large dataset of the flexibility of CDRs and related protein loops which collects the structures of all conformations observed in crystal structures. While some loops adopt multiple conformational states, others are always observed in an identical conformation (Figure 1, Panel A-B). Using the dataset, we develop AbFlex, a method that accurately predicts the flexibility of CDRs from inputs of either crystal structures or predicted structural models (Figure 1, Panel C).

---

[1]Department of Statistics, University of Oxford, Oxford, UK
[2]Large Molecule Research, Roche Pharma Research and Early Development, Roche Innovation Center Munich, Penzberg, Germany. Correspondence to: Charlotte M. Deane <deane@stats.ox.ac.uk>.

*Accepted at the 1st Machine Learning for Life and Material Sciences Workshop at ICML 2024.* Copyright 2024 by the author(s).

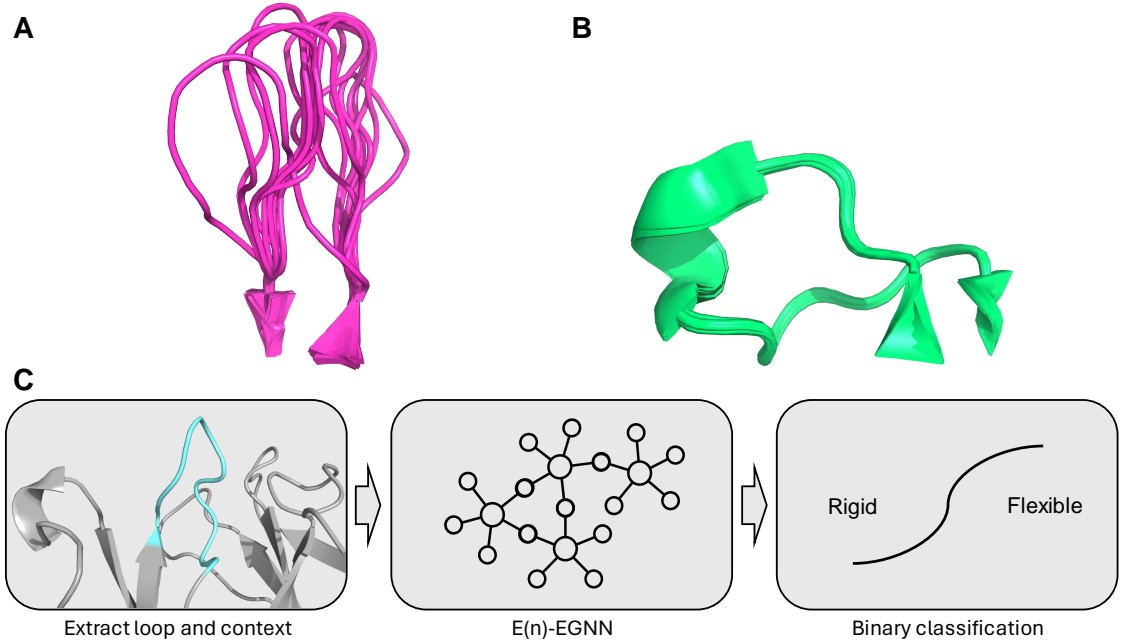

*Figure 1.* Overview of the AbFlex method. A) Example of a 'flexible' protein loop which is observed to adopt multiple conformations. 12 crystal structures of loops identical in sequence are overlaid. B) Example of a 'rigid' loop which adopts a single conformation. 14 crystal structures of sequence identical loops are overlaid. C) Flowchart detailing the AbFlex method predicting the conformational flexibility of CDR loops. The structure and sequence of a loop (cyan) and its context (grey) are extracted from a PDB file and a graph representation is generated. A three-layer E(n)-EGNN iteratively updates the node features, followed by binary classification of the loop as conformationally flexible or rigid.

## 2. Dataset

The amount of data on the conformational flexibility of proteins is limited which makes it difficult to train or even benchmark methods. Evidence on conformational flexibility can be obtained from crystal structures. Although X-ray crystallography does not directly capture molecular motions, low-energy conformations of a protein should appear in structures solved under different conditions. Therefore, determining structural flexibility from crystallographic data requires multiple solved structures of a protein which restricts the number of available data points as compared to standard protein structure prediction. Crystallographic data is often used for case studies but systematic mining of the PDB for instances of the same protein remains underexplored (Ellaway et al., 2023).

Molecular dynamics (MD) simulations provide an alternative to generate structures of conformational ensembles. MD simulations are computationally expensive and, therefore, even the largest databases of standardised MD simulations are not sufficient for training machine learning models (Vander Meersche et al., 2024). Furthermore, model assumptions, imperfections in force field parameterisations, sensitivity to initial conditions and accessible timescales

may lead to inaccuracies in simulated flexibility behaviours (Wan et al., 2021).

Here, we focus on evidence of conformational flexibility of antibody CDRs from crystallographic data and collect the structures of all observed conformations. To create a large dataset for training a robust machine learning model, we implement a systematic approach to mine all PDB structures for CDR-like secondary structure motifs. Motifs are defined as any loop connecting antiparallel $\beta$-strands. Furthermore, we create a high confidence test set of CDRH3 and CDRL3 flexibility by searching the Structural Antibody Database (SAbDab) (Schneider et al., 2022; Dunbar et al., 2014). Furthermore, data points are assigned binary labels. Loops are identified as 'flexible', if they adopt two or more distinct conformations, and 'rigid', if multiple identical conformations are observed and there is no evidence for multiple accessible conformations (Figure 1, Panel A-B).

### 2.1. Protein Loops

A large dataset of the conformational flexibility of CDR-like loops in general proteins was created as detailed below. All crystal structures deposited in the PDB with resolution under 3.5 Å were mined using DSSP, an algorithm for sec-

**Algorithm 1** Labelling of data points

**Require:** Loop sequences $s$, coordinates $x$, PDB code of structure containing loop $p$

  Group $x$ and $p$ pairs by $s$ identity:

  $group_i \leftarrow \{(x,p)_{l:m}\}_i$

  **for** $n = 0$ **to** $N$ **do**

    $alignedCoords \leftarrow \text{AlignCoordinates}(group_n.x)$

    $distanceMat \leftarrow \text{pairwiseRMSD}(alignedCoords)$

    **if** $\max(distanceMat) > 1.25$ Å **then**

      $label_n \leftarrow$ flexible

    **else if** $\text{len}(\text{set}(group_n.p)) \geq 5$ **then**

      $label_n \leftarrow$ rigid

    **else**

      $label_n \leftarrow$ ambiguous

    **end if**

  **end for**

ondary structure assignment (Kabsch & Sander, 1983), for loop motifs connecting antiparallel $\beta$-strands. 1,200,000 occurrences of protein loop structures were found. Structures were grouped by sequence identity and groups were labelled as conformationally 'flexible' or 'rigid' as outlined in Algorithm 1. Structures in each group are aligned on the C$\alpha$ of loop residues and a pairwise distance matrix of C$\alpha$ RMSDs was calculated. If the RMSD of any two structures in a group exceeded 1.25 Å the group was labelled to be 'flexible'. The cutoff of 1.25 Å was chosen as this is known to provide a good threshold for functional clustering of CDR loops (Spoendlin et al., 2023). The absence of evidence for flexibility does not prove that a loop cannot adopt multiple conformations, but the possibility that alternative states were not captured remains. The more structures depicting the loop to adopt a single conformation are available, the more confident we can be on the absence of flexibility. We, therefore, introduce a requirement that a loop needs to adopt a single conformation in at least five separate PDB structures to be labelled as 'rigid' [1]. The final dataset contains more than 20,000 labelled loops (Table 1).

## 2.2. Antibody CDRs

Additionally, a high confidence dataset of conformational flexibility of antibody CDRs was created. All fragment variable (Fv) structures were extracted from SAbDab (Schneider et al., 2022; Dunbar et al., 2014). Structures were grouped by Fv sequence identity, rather than CDR sequence identity, to increase confidence in the determined flexibility.

---

[1]We chose the requirement of five separate PDB files instead of simply five structures as the same loop can occur several times in the same PDB file due to multiple copies of a protein within a crystal unit cell. Loops are likely to adopt the same conformation in all copies even if flexible.

*Table 1.* Datasets and sizes.

| DATASET | TOTAL | FLEXIBLE | RIGID |
|---|---|---|---|
| PROTEIN LOOPS | 20,216 | 4289 | 15,927 |
| CDRH3 | 152 | 97 | 55 |
| CDRL3 | 84 | 15 | 69 |

CDRH3s and CDRL3[2] of each group were then labelled as 'flexible' or 'rigid' identical as for the protein loops dataset.

## 3. Method

In order to demonstrate the usefulness of the dataset we developed AbFlex, a model classifying the flexibility of CDR loops.

### 3.1. AbFlex model

AbFlex is a graph neural network and consists of three E(n)-EGNN layers (Satorras et al., 2022). The layers take input of node embeddings $h^l$, coordinates $x^l$ and edge embeddings $\mathcal{E} = a_{ij}$. Embeddings in each layers are updated as follows:

$$m_{ij} = \phi_m(h_i^l, h_j^l, \|x_i^l - x_j^l\|^2, a_{ij})$$

$$x_i^{l+1} = x_i^l + C \sum_{j \neq i} (x_i^l - x_j^l)\, \phi_x(m_{ij})$$

$$m_i = \sum_{j \neq i} m_{ij}$$

$$h_i^{l+1} = \phi_h(h_i^l, m_i)$$

The last layer of node features are pooled and a linear layer with sigmoid activation function applied for binary classification. The chosen model architecture makes predicted flexibilities invariant to transformations of the group E(3) (translations, rotations, reflections), therefore orientations and absolute positions of inputs to the model can be ignored.

A loop and its structural context are encoded as a residue level graph. Context is provided by all residues within 10 Å of any loop residues. Node features are a 22-dimensional vector formed by a one-hot encoding of amino acid type (1 class for each of the 20 amino acids plus an additional class for unknown residues) concatenated with a one-hot encoding of the residue being located in the loop or context. A 10 Å distance cutoff is used to construct the edges. Edge

---

[2]CDR3s are here defined as IMGT numbered (Lefranc et al., 2003) residues 107-116. In most antibodies these residues form the loop between the two antiparallel $\beta$-strands.

features are 9-dimensional providing a one-hot encoding of the presence of a covalent bond between two residues and a C$\alpha$ distance encoding. The distance encoding is produced by 8 Gaussian radial basis functions (RBFs) equally distributed between 0 Å and 10 Å. Residue coordinates were taken from the C$\alpha$ atoms.

A 70-15-15 split was used to divide the protein loops dataset into training, validation and test sets. A maximal sequence identity of 80% between the splits was enforced. Additionally, all loops with more than 80% sequence identity to any of the loops in the CDRH3 and CDRL3 datasets were removed from the training and validation sets. One structure per loop was sampled for the validation and test sets. To ensure the stability of predictions to small changes in atom coordinates the training set was augment by the random sampling of five structures per loop.

AbFlex was trained with a binary cross-entropy loss using the Adam optimiser (Kingma & Ba, 2014) with a learning rate of $2 \cdot 10^{-4}$ and weight decay of $10^{-6}$. The validation area under the precision-recall curve (PR AUC) was monitored and training stopped when converged.

### 3.2. Baseline models

A set of three baseline models were created to assess AbFlex performance. Firstly, a logistic regression classifier was fit to inputs of loop length. Longer loops contain more bonds around which they can rotate and are therefore expected to be more flexible in conformation (Guloglu & Deane, 2023). Secondly, a logistic regression classifier was fit to inputs of the solvent exposure of a loop. We approximate solvent exposure by the number of residues located within 10 Å around the loop. Loops with higher solvent exposure have less steric hindrance restricting conformational rearrangements and are expected to be more flexible (Guloglu & Deane, 2023). Lastly, a logistic regression was fit to both length and solvent exposure. All baselines were trained and tested on identical data splits as AbFlex.

### 3.3. AF2-based flexibility prediction

AbFlex performance on the CDR test sets was compared to two AF2-based approaches for flexibility prediction. Besides the predicted structures, AF2 returns a pLDDT score for each residues which provides a local confidence measure. Low pLDDT scores have previously been described to provide a good indicator that a part of a protein is disordered and flexible in conformation (Jumper et al., 2021). We, therefore, used the mean pLDDT of a CDR as a predictor of flexibility. Furthermore, AF2 workflows using subsampled, shallow MSAs can successfully predict conformational states for some proteins (Riccabona et al., 2024). We created 40 models of each antibody running AF2 with

MSA subsampling[3] and used the average CDR RMSD in the 40 models as a predictor of flexibility.

## 4. Results

### 4.1. Performance on protein loops dataset

AbFlex was initially evaluated on the protein loops test set (Table 2). To assess its performance, we compare AbFlex predictions to three simple baseline models. AbFlex achieves good predictive power at classifying loops to be 'rigid' or 'flexible' and outperforms all baselines substantially. Baseline results show that features of loop length and solvent exposure detect some signal, however are not sufficient to explain most of the loop flexibility. The superior performance of AbFlex suggests that the model has learned more than these two simple features to predict conformational flexibility.

Additionally, we tested two modifications of the AbFlex method. AbFlex-loop uses an identical architecture as AbFlex but model input differs. AbFlex-loop is given a graph encoding of only the loop itself and not its structural context. AbFlex-sequence is a CNN-based model trained on an amino acid sequence encoding of the loop (again no data of the structural context is provided). AbFlex-loop and AbFlex-sequence achieve similar performance (Table 2) indicating that loop structure alone does not give more information than sequence. Both models outperform the only relevant baseline (baseline-length)[4], but show less predictive power than AbFlex.

These results give some indication on the biophysical factors that affect the conformational flexibility of protein loop motifs. Long loops tend to be more flexible than shorter ones. Furthermore, the sequence of a loop impacts its ability to adopt multiple conformations; no additional information is gained from the structure of the loop alone. Encoding the structural context of the loop motif results in the largest boost in performance. This highlights that interactions with the context within the protein are an important factor affecting conformational dynamics.

### 4.2. Predicting CDR flexibility

AbFlex was further evaluated on the CDRH3 and CDRL3 test sets. The methods generalises to antibody CDRs and achieves good predictive power on both sets. AbFlex substantially outperforms all baseline methods and AF2-based

---

[3]The following parameters of AF2 were changed from default values: MSA depth = 64, extra sequences = 128, recycles = 1, seeds = 8. Parameters were selected to maximise the diversity of outputs while limiting the occurrence of unfolded structures.

[4]Baseline-length is the only relevant baseline here. The other two consider information of the loop's structural context which AbFlex-loop and AbFlex-sequence do not have access to.

*Table 2.* Protein loops test set performance.

| METHOD | ROC AUC | PR AUC |
|---|---|---|
| RANDOM | 0.50 | 0.23 |
| BASELINE - SOLVENT EXPOSURE | 0.60 | 0.29 |
| BASELINE - LENGTH | 0.69 | 0.35 |
| BASELINE - COMBINED | 0.76 | 0.46 |
| ABFLEX-SEQUENCE | 0.76 | 0.47 |
| ABFLEX-LOOP | 0.76 | 0.46 |
| ABFLEX | **0.83** | **0.61** |

*Table 3.* CDR test set performance. Values specify the area under the precision-recall curve (PR AUC).

| METHOD | CDRH3 | CDRL3 |
|---|---|---|
| RANDOM | 0.64 | 0.18 |
| BASELINE - SOLVENT EXPOSURE | 0.68 | 0.32 |
| BASELINE - LENGTH | 0.69 | 0.36 |
| BASELINE - COMBINED | 0.72 | 0.37 |
| ABFLEX | **0.80** | 0.62 |
| ABFLEX-ABB2 | 0.77 | **0.63** |
| AF2 - PLDDT | 0.70 | 0.32 |
| AF2 - MSA SUBSAMPLING | 0.74 | 0.48 |

alternatives.

The CDRH3 test set contains an interesting case study of a loop that is flexible in the context of one Fv sequence (A) but rigid in the context of an alternative Fv (B) (Figure 2). The two Fvs differ by a slight repositioning of a $\beta$-strand near the CDRH3. AbFlex predicts the increased flexibility of A (prediction score 0.41) compared to B (0.04). This case study indicates that AbFlex is sensitive to small rearrangements in the structural context that affect CDR flexibility.

Up to this points, AbFlex was only tested with structural data taken from crystal structures. Sequence data is available for a far grater number of antibodies (Olsen et al., 2022) than experimentally solved structures (Dunbar et al., 2014). We, therefore, tested AbFlex on input graphs created from structural models predicted using ABodyBuilder2 (Abanades et al., 2023). Similar performance is achieved using predicted structures (AbFlex-ABB2 in Table 3) as for crystal structures (AbFlex in Table 3), showing that AbFlex can be used even if only antibody sequence is known.

## 5. Conclusions

Conformational changes give rise to functional properties of many classes of proteins (Teilum et al., 2009). Current machine learning tools do not capture structural flexibility well (Riccabona et al., 2024; Jing et al., 2024). A main factor that has limited methods development in this space is the absence of large datasets necessary to train and evaluate models. Here, we focus on the conformational flexibility of antibody CDRs, a functionally highly important protein motif. We mine the PDB (Berman et al., 2002) and SAbDab (Dunbar et al., 2014) for crystallographic evidence of the conformational flexibility of CDRs and structurally related loops across all classes of proteins. Through this approach we create a large dataset set of more than 20,000 loop motifs with determined flexibility.

We develop AbFlex which shows strong predictive power for classifying if antibody CDRs are able to transition between multiple conformational states or consistently adopt a single stable conformation. Our method substantially outperforms AF2-based alternative which have previously been described as predictors of protein flexibility (Jumper et al., 2021; del Alamo et al., 2022). By training on crystal structure data, we eliminate potential artefacts originating from methods, e.g. MD, that approximate the flexibility of proteins through simulation. The conformational flexibility of CDRs affects functional properties of the antibody including affinity (Mikolajek et al., 2022) and specificity (Guthmiller et al., 2020; James et al., 2003), which are key properties that need to be optimise in therapeutic drugs. AbFlex, therefore, adds a valuable tool to investigate antibody function and assist the drug discovery process.

Furthermore, this work highlights biophysical factors that influence the conformational flexibility of protein loop motifs. While sequence affects the tendency to adopt multiple conformations, we identify the arrangement of residues in the surrounding (structural context) as a key factor that drives loop flexibility. These results are in line with previous studies of CDRH3 flexibility (Guloglu & Deane, 2023) and suggest that this finding may generalise to all protein loop motifs.

## Software and data availability

The AbFlex code and training data will be made available upon publication.

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

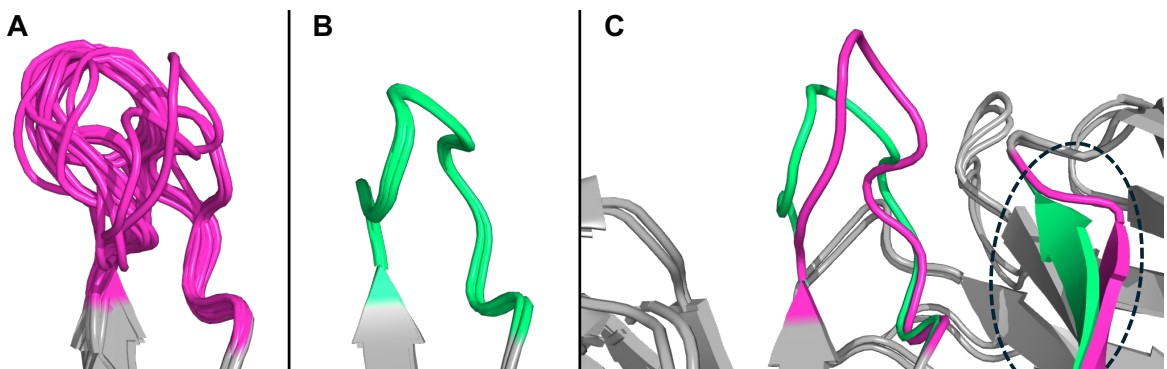

*Figure 2.* Case study of CDRH3 conformational flexibility. The CDRH3 sequence WGGDGFYAMD appears in the context of two Fv sequences. A) 11 PDB structures of the CDR (magenta) in context A show several distinct conformations. B) 8 PDB structures of the CDR (lime) provide evidence for a single conformation in context B. C) The Fvs sequences differ in several residues at the interface of the heavy and light chain which leads to a rearrangement of a $\beta$-strand (circled) in the proximity of CDRH3. The slight bend in the $\beta$-strand in context A (magenta) leaves more space to the CDRH3 which likely allows the loop to access the alternative conformations.

L. N., Schaeffer, R. D., Millán, C., Park, H., Adams, C., Glassman, C. R., DeGiovanni, A., Pereira, J. H., Rodrigues, A. V., van Dijk, A. A., Ebrecht, A. C., Opperman, D. J., Sagmeister, T., Buhlheller, C., Pavkov-Keller, T., Rathinaswamy, M. K., Dalwadi, U., Yip, C. K., Burke, J. E., Garcia, K. C., Grishin, N. V., Adams, P. D., Read, R. J., and Baker, D. Accurate prediction of protein structures and interactions using a three-track neural network. *Science*, 373(6557):871–876, August 2021. ISSN 0036-8075, 1095-9203. doi: 10.1126/science. abj8754. URL https://www.science.org/doi/ 10.1126/science.abj8754.

Berman, H. M., Battistuz, T., Bhat, T. N., Bluhm, W. F., Bourne, P. E., Burkhardt, K., Feng, Z., Gilliland, G. L., Iype, L., Jain, S., Fagan, P., Marvin, J., Padilla, D., Ravichandran, V., Schneider, B., Thanki, N., Weissig, H., Westbrook, J. D., and Zardecki, C. The Protein Data Bank. *Acta Crystallographica Section D Biological Crystallography*, 58(6):899–907, June 2002. ISSN 0907-4449. doi: 10.1107/S0907444902003451. URL https://scripts.iucr.org/cgi-bin/ paper?S0907444902003451.

del Alamo, D., Sala, D., Mchaourab, H. S., and Meiler, J. Sampling alternative conformational states of transporters and receptors with AlphaFold2. *eLife*, 11:e75751, March 2022. ISSN 2050-084X. doi: 10.7554/eLife.75751. URL https://doi.org/10.7554/eLife.75751.

Dunbar, J., Krawczyk, K., Leem, J., Baker, T., Fuchs, A., Georges, G., Shi, J., and Deane, C. M. SAbDab: the structural antibody database. *Nucleic Acids Research*, 42 (D1):D1140–D1146, January 2014. ISSN 0305-1048, 1362-4962. doi: 10.1093/nar/gkt1043. URL https://

academic.oup.com/nar/article-lookup/ doi/10.1093/nar/gkt1043.

Ellaway, J. I. J., Anyango, S., Nair, S., Zaki, H. A., Nadzirin, N., Powell, H. R., Gutmanas, A., Varadi, M., and Velankar, S. Automated Pipeline for Comparing Protein Conformational States in the PDB to AlphaFold2 Predictions, July 2023. URL http://biorxiv.org/ lookup/doi/10.1101/2023.07.13.545008.

Faezov, B. and Dunbrack, R. L. AlphaFold2 models of the active form of all 437 catalytically-competent typical human kinase domains, July 2023. URL https://www.biorxiv.org/content/10. 1101/2023.07.21.550125v1.

Guloglu, B. and Deane, C. M. Specific attributes of the VL domain influence both the structure and structural variability of CDR-H3 through steric effects. *Frontiers in Immunology*, 14:1223802, July 2023. ISSN 1664-3224. doi: 10.3389/fimmu.2023.1223802. URL https://www.frontiersin.org/articles/ 10.3389/fimmu.2023.1223802/full.

Guthmiller, J. J., Lan, L. Y.-L., Fernández-Quintero, M. L., Han, J., Utset, H. A., Bitar, D. J., Hamel, N. J., Stovicek, O., Li, L., Tepora, M., Henry, C., Neu, K. E., Dugan, H. L., Borowska, M. T., Chen, Y.-Q., Liu, S. T. H., Stamper, C. T., Zheng, N.-Y., Huang, M., Palm, A.-K. E., García-Sastre, A., Nachbagauer, R., Palese, P., Coughlan, L., Krammer, F., Ward, A. B., Liedl, K. R., and Wilson, P. C. Polyreactive Broadly Neutralizing B cells Are Selected to Provide Defense against Pandemic Threat Influenza Viruses. *Immunity*, 53(6): 1230–1244.e5, December 2020. ISSN 1097-4180. doi: 10.1016/j.immuni.2020.10.005.

James, L. C., Roversi, P., and Tawfik, D. S. Antibody Multispecificity Mediated by Conformational Diversity. *Science*, 299(5611):1362–1367, February 2003. ISSN 0036-8075, 1095-9203. doi: 10.1126/science.1079731. URL https://www.science.org/doi/10.1126/science.1079731.

Jing, B., Erives, E., Pao-Huang, P., Corso, G., Berger, B., and Jaakkola, T. EigenFold: Generative Protein Structure Prediction with Diffusion Models, April 2023. URL http://arxiv.org/abs/2304.02198. arXiv:2304.02198 [physics, q-bio].

Jing, B., Berger, B., and Jaakkola, T. AlphaFold Meets Flow Matching for Generating Protein Ensembles, February 2024. URL http://arxiv.org/abs/2402.04845. arXiv:2402.04845 [cs, q-bio].

Jumper, J., Evans, R., Pritzel, A., Green, T., Figurnov, M., Ronneberger, O., Tunyasuvunakool, K., Bates, R., Žídek, A., Potapenko, A., Bridgland, A., Meyer, C., Kohl, S. A. A., Ballard, A. J., Cowie, A., Romera-Paredes, B., Nikolov, S., Jain, R., Adler, J., Back, T., Petersen, S., Reiman, D., Clancy, E., Zielinski, M., Steinegger, M., Pacholska, M., Berghammer, T., Bodenstein, S., Silver, D., Vinyals, O., Senior, A. W., Kavukcuoglu, K., Kohli, P., and Hassabis, D. Highly accurate protein structure prediction with AlphaFold. *Nature*, 596(7873):583–589, August 2021. ISSN 1476-4687. doi: 10.1038/s41586-021-03819-2. URL https://www.nature.com/articles/s41586-021-03819-2.

Kabsch, W. and Sander, C. Dictionary of protein secondary structure: pattern recognition of hydrogen-bonded and geometrical features. *Biopolymers*, 22(12):2577–2637, December 1983. ISSN 0006-3525. doi: 10.1002/bip.360221211. URL https://onlinelibrary.wiley.com/doi/10.1002/bip.360221211.

Kingma, D. P. and Ba, J. Adam: A Method for Stochastic Optimization, 2014. URL https://arxiv.org/abs/1412.6980.

Lefranc, M.-P., Pommie, C., Ruiz, M., Giudicelli, V., Foulquier, E., Truong, L., Thouvenin-Contet, V., and Lefranc, G. IMGT unique numbering for immunoglobulin and T cell receptor variable domains and Ig superfamily V-like domains. *Developmental & Comparative Immunology*, 27(1):55–77, January 2003. ISSN 0145-305X. doi: 10.1016/S0145-305X(02)00039-3. URL https://www.sciencedirect.com/science/article/pii/S0145305X02000393.

Lin, Z., Akin, H., Rao, R., Hie, B., Zhu, Z., Lu, W., Smetanin, N., Verkuil, R., Kabeli, O., Shmueli, Y., Dos Santos Costa, A., Fazel-Zarandi, M., Sercu, T., Candido, S., and Rives, A. Evolutionary-scale prediction of atomic-level protein structure with a language model. *Science*, 379(6637):1123–1130, March 2023. ISSN 0036-8075, 1095-9203. doi: 10.1126/science.ade2574. URL https://www.science.org/doi/10.1126/science.ade2574.

Lu, J., Zhong, B., and Tang, J. Score-based Enhanced Sampling for Protein Molecular Dynamics. July 2023. URL https://openreview.net/forum?id=NO3QwxuHv9.

Mansoor, S., Baek, M., Park, H., Lee, G. R., and Baker, D. Protein Ensemble Generation Through Variational Autoencoder Latent Space Sampling. *Journal of Chemical Theory and Computation*, 20(7):2689–2695, April 2024. ISSN 1549-9618, 1549-9626. doi: 10.1021/acs.jctc.3c01057. URL https://pubs.acs.org/doi/10.1021/acs.jctc.3c01057.

Mikolajek, H., Weckener, M., Brotzakis, Z. F., Huo, J., Dalietou, E. V., Le Bas, A., Sormanni, P., Harrison, P. J., Ward, P. N., Truong, S., Moynie, L., Clare, D. K., Dumoux, M., Dormon, J., Norman, C., Hussain, N., Vogirala, V., Owens, R. J., Vendruscolo, M., and Naismith, J. H. Correlation between the binding affinity and the conformational entropy of nanobody SARS-CoV-2 spike protein complexes. *Proceedings of the National Academy of Sciences*, 119(31):e2205412119, August 2022. ISSN 0027-8424, 1091-6490. doi: 10.1073/pnas.2205412119. URL https://pnas.org/doi/full/10.1073/pnas.2205412119.

Olsen, T. H., Boyles, F., and Deane, C. M. Observed Antibody Space: A diverse database of cleaned, annotated, and translated unpaired and paired antibody sequences. *Protein Science*, 31(1):141–146, January 2022. ISSN 0961-8368, 1469-896X. doi: 10.1002/pro.4205. URL https://onlinelibrary.wiley.com/doi/10.1002/pro.4205.

Riccabona, J. R., Spoendlin, F. C., Fischer, A.-L. M., Loeffler, J. R., Quoika, P. K., Jenkins, T. P., Ferguson, J. A., Smorodina, E., Laustsen, A. H., Greiff, V., Forli, S., Ward, A., Deane, C. M., and Fernandez-Quintero, M. L. Assessing AF2's ability to predict structural ensembles of proteins, April 2024. URL http://biorxiv.org/lookup/doi/10.1101/2024.04.16.589792.

Sala, D., Engelberger, F., Mchaourab, H. S., and Meiler, J. Modeling conformational states of proteins with AlphaFold. *Current Opinion in Structural Biology*, 81:102645, August 2023a. ISSN 0959-440X. doi: 10.1016/j.sbi.2023.102645. URL https://www.sciencedirect.com/science/article/pii/S0959440X23001197.

Sala, D., Hildebrand, P. W., and Meiler, J. Biasing AlphaFold2 to predict GPCRs and kinases with user-defined functional or structural properties. *Frontiers in Molecular Biosciences*, 10, 2023b. ISSN 2296-889X. doi: 10.3389/fmolb.2023.1121962. URL https://www.frontiersin.org/articles/10.3389/fmolb.2023.1121962.

Saldaño, T., Escobedo, N., Marchetti, J., Zea, D. J., Mac Donagh, J., Velez Rueda, A. J., Gonik, E., García Melani, A., Novomisky Nechcoff, J., Salas, M. N., Peters, T., Demitroff, N., Fernandez Alberti, S., Palopoli, N., Fornasari, M. S., and Parisi, G. Impact of protein conformational diversity on AlphaFold predictions. *Bioinformatics*, 38(10): 2742–2748, May 2022. ISSN 1367-4803, 1367-4811. doi: 10.1093/bioinformatics/btac202. URL https://academic.oup.com/bioinformatics/article/38/10/2742/6563595.

Satorras, V. G., Hoogeboom, E., and Welling, M. E(n) Equivariant Graph Neural Networks, February 2022. URL http://arxiv.org/abs/2102.09844. arXiv:2102.09844 [cs, stat].

Schneider, C., Raybould, M. I. J., and Deane, C. M. SAbDab in the age of biotherapeutics: updates including SAbDab-nano, the nanobody structure tracker. *Nucleic Acids Research*, 50(D1):D1368–D1372, January 2022. ISSN 0305-1048, 1362-4962. doi: 10.1093/nar/gkab1050. URL https://academic.oup.com/nar/article/50/D1/D1368/6431822.

Spoendlin, F. C., Abanades, B., Raybould, M. I. J., Wong, W. K., Georges, G., and Deane, C. M. Improved computational epitope profiling using structural models identifies a broader diversity of antibodies that bind to the same epitope. *Frontiers in Molecular Biosciences*, 10, 2023. ISSN 2296-889X. doi: 10.3389/fmolb.2023.1237621. URL https://www.frontiersin.org/articles/10.3389/fmolb.2023.1237621.

Stein, R. A. and Mchaourab, H. S. SPEACH_af: Sampling protein ensembles and conformational heterogeneity with Alphafold2. *PLOS Computational Biology*, 18(8):e1010483, August 2022. ISSN 1553-7358. doi: 10.1371/journal.pcbi.1010483. URL https://journals.plos.org/ploscompbiol/article?id=10.1371/journal.pcbi.1010483.

Teilum, K., Olsen, J. G., and Kragelund, B. B. Functional aspects of protein flexibility. *Cellular and Molecular Life Sciences*, 66(14):2231–2247, July 2009. ISSN 1420-9071. doi: 10.1007/s00018-009-0014-6. URL https://doi.org/10.1007/s00018-009-0014-6.

Vander Meersche, Y., Cretin, G., Gheeraert, A., Gelly, J.-C., and Galochkina, T. ATLAS: protein flexibility description from atomistic molecular dynamics simulations. *Nucleic Acids Research*, 52(D1):D384–D392, January 2024. ISSN 0305-1048, 1362-4962. doi: 10.1093/nar/gkad1084. URL https://academic.oup.com/nar/article/52/D1/D384/7438909.

Wan, S., Sinclair, R. C., and Coveney, P. V. Uncertainty quantification in classical molecular dynamics. *Philosophical Transactions of the Royal Society A: Mathematical, Physical and Engineering Sciences*, 379(2197), May 2021. ISSN 1364-503X, 1471-2962. doi: 10.1098/rsta.2020.0082. URL https://royalsocietypublishing.org/doi/10.1098/rsta.2020.0082.

Wayment-Steele, H. K., Ojoawo, A., Otten, R., Apitz, J. M., Pitsawong, W., Hömberger, M., Ovchinnikov, S., Colwell, L., and Kern, D. Predicting multiple conformations via sequence clustering and AlphaFold2. *Nature*, 625(7996): 832–839, January 2024. ISSN 1476-4687. doi: 10.1038/s41586-023-06832-9. URL https://www.nature.com/articles/s41586-023-06832-9.

Zheng, S., He, J., Liu, C., Shi, Y., Lu, Z., Feng, W., Ju, F., Wang, J., Zhu, J., Min, Y., Zhang, H., Tang, S., Hao, H., Jin, P., Chen, C., Noé, F., Liu, H., and Liu, T.-Y. Towards Predicting Equilibrium Distributions for Molecular Systems with Deep Learning, 2023. URL https://arxiv.org/abs/2306.05445.