# OpenReview forum: "AbFlex: Predicting the conformational flexibility of antibody CDRs"
_ICML.cc/2024/Workshop/ML4LMS — ML4LMS Poster_

### Official Review · Reviewer_Ynvz · 2024-06-02
**AbFlex is a compelling model for binary predictions of loop conformational flexibility, which is an important consideration for tasks such as antibody structure prediction and design. AbFlex is of value to the community.**

**Rating:** 8
**Confidence:** 5

**Review:**

AbFlex is a compelling model for binary predictions of loop conformational flexibility, which is an important consideration for tasks such as antibody structure prediction and design. Such predictions could be valuable for contextualizing the outputs of models such as ImmuneBuilder. Furthermore, the "Protein Loops" dataset the authors create could be useful as a pretraining dataset for other tasks. The ablations and baselines are valuable for understanding the usefulness of the model and the trastuzumab HCDR3 case study shows promising evidence of AbFlex's ability to use information in the antibody framework structures. I believe the paper is well written and would be of value to the community. I list specific pros and cons below with an emphasis towards the latter with the goal of providing suggestions for improving the paper.

Pros:
* The task AbFlex is trying to solve is clearly laid out and an important problem.
* The dataset construction for "Protein Loops" is motivated effectively and described well.
* The authors include baselines using biophysical properties such as solvent exposure and loop length and show their model, when provided the full structure, is superior. They also show that including the full structure is valuable compared to a sequence-only approach and a loop-only structure approach.
* The authors evaluate AbFlex on predicted structures from ABodyBuilder2 and find competitive performance to solved structures, suggesting their model can be used in practical settings with modeled structures.

Cons:
* One thing that confused me regarding the training procedure that I would appreciate clarification on is how loops were sub-sampled. The authors write
```
One structure per loop was sampled for the validation and test sets. To
ensure the stability of predictions to small changes in atom
coordinates the training set was augment by the sampling of
five structures per loop.
```
I would like to know how this sampling was done. Was it random or done using something like a structure quality or loop diversity metric?
* The combined baseline is competitive with the ablated models, which suggests sequence-only or loop-only are not that powerful. Did the authors consider adding solvent exposure as a feature to AbFlex? Perhaps that would boost performance.
* An MD-like baseline is missing and would be appreciated. While running MD would be quite expensive, perhaps prohibitively so, something like change in loop structure after an OpenMM minimization might be relevant.
* A model such as ABodyBuilder2 may have low confidence in its predictions when there is significant conformational flexibility. A baseline using ABodyBuilder2 pRMSD or AlphaFold2/ESMFold pLDDT would be interesting as well.
* Have the authors tried modeling the scalar RMSD values as opposed to a binary label of rigid vs. flexible? Or using a set of bins such as "rigid", "low flexibility", "flexible", "high flexibility"?
* One item of work in the literature that I believe the authors missed on CDR conformational flexibility is a recent work from Andrew Martin's lab titled "Do antibody CDR loops change conformation upon binding?" (https://www.tandfonline.com/doi/epdf/10.1080/19420862.2024.2322533?needAccess=true). This work finds that most antibody loops, including HCDR3, do not change conformations upon binding. They curate a dataset of bound and unbound solved antibody structures to verify this. Evaluating AbFlex on this dataset would be quite interesting, as ultimately we care about antibodies in the context of their binding to targets.

---

### Official Review · Reviewer_qN11 · 2024-06-11
**A new computational strategy to better predict antibody loop flexibility**

**Rating:** 9
**Confidence:** 5

**Review:**

1. Quality: This is high-quality work, capturing useful strategies in the field of predicting antibody loop fexability
2. Clarity: well-written text
3. Originality: this article makes new headway for antibodies and will be relevant to other protein/peptide formats.
4. Significance of this work: antibody CDR3 flexibility is a real challenge in drug discovery and papers provides a new route to address this.
5. Pros Significant method development with clear examples comparing architectures ( regression, graph, NN) as well as data sources of sequence and structure on model performance.
6. Cons No obvious weakness

---

### Official Review · Reviewer_PeBb · 2024-06-12
**Review for: AbFlex: Predicting the conformational flexibility of antibody CDRs**

**Rating:** 5
**Confidence:** 4

**Review:**

-